# Multivariate Interaction Analysis of *Zea mays* L. Genotypes Growth Productivity in Different Environmental Conditions

**DOI:** 10.3390/plants12112165

**Published:** 2023-05-30

**Authors:** Nataša Ljubičić, Vera Popović, Marko Kostić, Miloš Pajić, Maša Buđen, Kosta Gligorević, Milan Dražić, Milica Bižić, Vladimir Crnojević

**Affiliations:** 1BioSense Institute, University of Novi Sad, 21000 Novi Sad, Serbia; natasa.ljubicic@biosense.rs (N.L.); masa.budjen@biosense.rs (M.B.); crnojevic@biosense.rs (V.C.); 2Institute of Field and Vegetable Crops, 21000 Novi Sad, Serbia; 3Faculty of Agriculture, University of Bijeljina, 76300 Bijeljina, Bosnia and Herzegovina; 4Faculty of Agriculture, University of Novi Sad, 21000 Novi Sad, Serbia; marko.kostic@polj.uns.ac.rs; 5Faculty of Agriculture, University of Belgrade, Nemanjina 6, 11000 Belgrade, Serbia; paja@agrif.bg.ac.rs (M.P.); koleg@agrif.bg.ac.rs (K.G.); mdrazic@agrif.bg.ac.rs (M.D.); milica.bizic8@gmail.com (M.B.)

**Keywords:** maize, grain yield productivity, genotype by environment interaction

## Abstract

Evaluating maize genotypes under different conditions is important for identifying which genotypes combine stability with high yield potential. The aim of this study was to assess stability and the effect of the genotype–environment interaction (GEI) on the grain yield traits of four maize genotypes grown in field trials; one control trial without nitrogen, and three applying different levels of nitrogen (0, 70, 140, and 210 kg ha^−1^, respectively). Across two growing seasons, both the phenotypic variability and GEI for yield traits over four maize genotypes (P0725, P9889, P9757 and P9074) grown in four different fertilization treatments were studied. The additive main effects and multiplicative interaction (AMMI) models were used to estimate the GEI. The results revealed that genotype and environmental effects, such as the GEI effect, significantly influenced yield, as well as revealing that maize genotypes responded differently to different conditions and fertilization measures. An analysis of the GEI using the IPCA (interaction principal components) analysis method showed the statistical significance of the first source of variation, IPCA1. As the main component, IPCA1 explained 74.6% of GEI variation in maize yield. Genotype G3, with a mean grain yield of 10.6 t ha^−1^, was found to be the most stable and adaptable to all environments in both seasons, while genotype G1 was found to be unstable, following its specific adaptation to the environments.

## 1. Introduction

Maize (*Zea mays* L.) is a significant cereal crop and has the most abundant worldwide production, supplying human nutrition, animal feed, and feedstock for many industrial products and biofuels [1]. In addition to outstanding progress in the selection of new maize genotypes, the potential of maize yield is largely affected by abiotic and biotic factors, themselves determined by a certain combination of different factors, such as soil and climatic conditions, solar radiation, quality of seed, genetic performance, the potential for hybrids, and field management practices [2,3,4]. Since it is projected that the world population will increase to nine billion people by the year 2050, certain estimates indicate that agricultural production will have to increase by at least 70% [5,6]. Globally, with the genetic improvement of new maize genotypes (followed by improvements in appropriate agronomic technology and practices), grain yield increased by an average of 111 kg ha^−1^ per year between 1965 and 2012 [7], with 60% of this improvement having been achieved with regard to production based on hybrids [8]. Over the past century, it has been shown that the use of mineral fertilizer has drastically increased the grain yield of maize. In maize production, the maize yield quantity and quality largely depend on nitrogen (N) management. In order to achieve optimum maize growth, photosynthesis, grain formation, and protein accumulation during the growing season, N is prerequisite. Considering that maize crops have higher N demand in relation to soils supply, N fertilizer application is common practice. However, soil N availability is not constant, and its amount depends on different factors, such as a water status, the type of soil, and climatic conditions [9,10]. Climatic conditions and N fertilizer application significantly affect the maize yield productivity. Maize is sensitive to different levels of N availability, showing a positive correlation between maize yields and in-season fertilizer nitrogen uptake [11]. However, the application of sufficient N fertilizer levels can cause environmental pollution. Therefore, improvement of important grain yield traits is still a promising way to improve maize production [12]. Despite the quantity of maize yield and the balanced use of nitrogen fertilizers, one of the main goals of maize breeders is to produce maize genotypes with high grain yields and stable reactions in different environments, as well as in conditions of low soil fertility and low fertilization supply. Recently, maize producers have been more interested in maize hybrids that produce stable yields under various environments, and breeders have made efforts to achieve these requirements. Therefore, knowledge regarding the genotype–environment interaction (GEI) and stability is of paramount importance, both for maize breeders and maize producers [13]. The wide adaptation of maize genotypes demonstrates the genotypes’ ability to achieve the production of stable grain yields across a wide range of agroecological environments in different growing regions [14,15,16]. The grain yield of maize is the most important, complex, and polygenic trait, and is often affected by genetic (G) and environmental factors (E) and their interaction (GEI). During plant development, some maize genotypes can be steady in different agroecological conditions, while certain genotypes can express differences in various growing conditions [17,18]. Maize genotypes have different reactions in different years and to different treatments, as well as in the combination of these factors, due to GEI. Genotype–environment interactions present the different genotypic responses of plants expressed across various environments [19,20]. The assessment stability of maize yield (and yield-related traits) provides valuable information about their behavior in different environments. Differences in the genotypes’ performances in different environments are a consequence of the effect of the GEI [21]. It has been confirmed that GEIs significantly contribute to the yield (and yield-related traits) of maize [22,23]. The GEI reflects the different genotypes’ responses to their environmental conditions; such differences make both the recommendation of genotypes for specific conditions, as well as the management of mineral nutrition, difficult. In both our decision-making processes and the determination of the ideal growing environments for maize hybrids, both the genotype and the GEI effects are very important since they provide a better exploitation of yield potential for specific environments and adapted maize genotypes [24]. In order to both assess the GEI effects of maize and identify its appropriate growing environments, different statistical analyses and methods have been developed. Among the methods proposed for the evaluation of the GEI in various crops, the additive main effect and multiplicative interaction (AMMI) model is considered to be the most appropriate and is the most widely used. The greatest contribution of this analysis lies in the ability to assess the structure of interactions between genotypes and environments and reduce different limitations in various crops [20]. The additive main effects and multiplicative interaction (AMMI) model combines and considers together the analysis of variance (ANOVA) and the principal component analysis (PCA) in a single model [25]. In the AMMI model, an analysis of variance serves to divide the variation into genotype main effects (G), environment main effects (E) and GEI; afterwards, it applies principal component analysis (PCA) to analyze the residual GEI effect [20,25]. During the last two decades, the application of the AMMI model to yield trials has been frequently used, and there have been several recent reports [26,27,28]. An understanding of the GEI factor could contribute to the identification of optimal growing conditions for hybrid assessments and recommendations for more adapted hybrids [29,30,31,32,33,34,35,36,37,38,39,40,41,42,43,44]. Since analysis provides information in the identification and recommendation of maize genotypes with high productivity and adaptation to environmental conditions, from the maize breeding and producer’s view it is important to estimate a genotype’s reaction to different nitrogen treatments. Therefore, the objective of the present investigation was to (1) examine the response of four maize genotypes to various N conditions; (2) estimate the effects of the genotype, environment, and their interaction on three important yield traits of maize genotypes using AMMI analysis; (3) select a genotype with a stable reaction across each treatment; and (4) identify appropriate growing environments.

The results of this investigation could provide a better understanding of the maize response and yield adaptation in various growing conditions, which is significant for genotype selection, recommendations, management practices, and yield production.

## 2. Results and Discussion

Maize grain yield is a complex trait, influenced by genetic and environmental factors, and their interaction, as well as by agronomic and management factors. Despite the fact that the genotype greatly affects the final grain yield and yield-related traits, the temperature, available moisture, and climatic conditions throughout growing season and plant development, especially during the important physiological stage, play an important role [44]. Since grain yield is the result of many quantitative traits controlled by numerous genes with small effects, maize grain yield can be increased. Individual yield traits can vary in various growing conditions, and by improving direct and indirect yield traits it is possible to increase grain yield [45]. In order to define and establish appropriate selection criteria for breeding programs, it is necessary to examine yield traits, their contribution, and their relation. The identification of yield components of maize genotypes, analysis of GEI models, and yield stability estimation are very important for the selection of high yielding and stable genotypes. On the assumption that GEI is not present, the variety trials only need to be conducted in more location [18]. Therefore, since that investigation in one environment could not provide an adequate estimation, investigations in various agro-ecological conditions provide additional information, which is enough to make the right decisions [46].

### 2.1. Maize Grain Yield

Grain yield as the final development of a plant is a very important component that decides the final yield amount [47,48].

The present results indicated that the average values of the grain yield of maize genotypes ranged within different environments between 8.13 t ha^−1^ in E5 (control—without nitrogen fertilization) and 15.32 t ha^−1^ in E4 (nitrogen application of 140 kg N ha^−1^) for the both growing seasons. In the first growing season of 2021, the average values of grain yield within different environments ranged between 10.10 t ha^−1^ in E1 (control—without nitrogen fertilization) and 15.32 t ha^−1^ in E4 (nitrogen application of 140 kg N ha^−1^). In the growing season of 2022, within different environments, the mean values of grain yield of maize ranged between 8.68 t ha^−1^ in E5 (control—without nitrogen fertilization) and 10.27 t ha^−1^ in E7 (nitrogen application of 140 kg N ha^−1^). The average values of grain yield among different genotypes ranged from 10.55 t ha^−1^ (G3) to 12.05 t ha^−1^ (G1) within different maize genotypes (Table 1).

In the control environment (E1) without nitrogen fertilizations, the greatest average value for grain yield was observed for maize genotype G4 in both seasons, having values of 11.15 t ha^−1^ in the first season and 9.45 t ha^−1^ in the second season. These results suggest that this genotype exposes a positive response with stable reaction in less favorable conditions, such as control conditions without nitrogen fertilizations. For the control variants, the lowest values were denoted for maize genotype G2 (8.69 t ha^−1^) in the first season and G1 (7.95 t ha^−1^) in the second growing season.

In the second environment, E2 (nitrogen application of 70 kg ha^−1^), in the first growing season, the greatest overall average values for grain yield were observed for maize genotypes G2 (16.69 t ha^−1^), G3 (14.39 t ha^−1^), G1 (12.01 t ha^−1^), and G4 (11.31 t ha^−1^). During the second vegetation season, for the nitrogen application of 70 kg N ha^−1^ (E6), the highest average mean values for grain yield were observed for maize genotypes G1 (10.68 t ha^−1^), G4 (9.71 t ha^−1^), G2 (9.22 t ha^−1^), and G3 (8.96 t ha^−1^). In the E3 environment, for the nitrogen application of 140 kg N ha^−1^, in the first growing season, the highest mean values for grain yield were denoted for the maize genotypes G2 (17.30 t ha^−1^), G1 (14.53 t ha^−1^), G3 (12.91 t ha^−1^), and G1 (10.27 t ha^−1^). In the second growing season, with the same level of nitrogen fertilization (E7), the greatest average values of grain yield were estimated for maize genotypes G1 (12.67 t ha^−1^), G4 (11.32 t ha^−1^), G3 (8.81 t ha^−1^), and G2 (8.30 t ha^−1^), as seen in Table 1. In the first growing season in the E4 environment, for the nitrogen application of 70 kg N ha^−1^, the greatest average for grain yield was estimated for maize varieties G1 (15.32 t ha^−1^), G2 (14.71 t ha^−1^), G3 (12.39 t ha^−1^), and G4 (12.13 t ha^−1^). During the second growing season in the E8 environment, for the treatment with 210 kg N ha^−1^ nitrogen application, the highest average was observed for maize genotypes G1 (12.36 t ha^−1^), G4 (10.15 t ha^−1^), G2 (8.36 t ha^−1^), and G3 (8.05 t ha^−1^). The greatest values of variance for the maize yield were observed in the first growing season in the E4 environment (8.68), while the lowest was in the E1 environment. In the second growing season, observed values of variance were lower in the E5 environment, while the greatest values were observed in the E7 environment. The standard deviation for grain yield varied from 0.75 in the E5 (control) environment to 2.95 in the E3 environment (at the nitrogen application of 140 kg N ha^−1^), as can be seen in Table 1. According to the AMMI stability value (ASV ranking) for the grain yield, maize genotype G3 had the lowest value, which shows that it was the most stable, while G2 with the greatest values was unstable.

The observed results indicated that different treatments affect the differences in grain yield of maize. In addition to the effect of the applied treatments, across both studied years, the climatic conditions, temperature, and precipitation amount were different, which influenced the effect of fertilization, and each maize genotype expressed a specific response to environmental conditions. In the first vegetation season, the weather conditions were more favorable and greater mean values were observed. A greater response of grain yield to the climatic conditions was estimated in the second growing season because the overall yield mean values were the lowest in the control, and higher within all levels of the nitrogen fertilization treatment. Furthermore, weather conditions such as a heat and drought had a significant influence on hybrids of earlier maturity groups. During the second growing season, with the extreme drought conditions that were present during the critical stages of maize growth, the differences were significantly increased, and the effects of the applied measures were more evident (Table 2). In general, late-maturing hybrids expressed higher productivity in the more favorable 2021 growing season than in the 2022 growing season. The negative impact of heat on maize grain yields during the critical grain-filling stages of maize across different FAO maturity groups has been observed in several studies [49,50]. According to Khan et al. [51], high temperature during critical stages has a negative impact and gives rise to low grain weight. Apart from the climatic conditions, the investigated maize genotypes expressed different responses to different amounts of N fertilization. The grain yield of certain maize genotypes indicated significant variations over different fertilization treatments. In general, in the first growing season, generally higher average values of the maize yield were observed compared with the second vegetation period. In this sense, genotype G1 showed a greater sensitivity to the applied treatments, as well as a positive response within each level of nitrogen application in the first growing season. The greatest values were observed in the E4 environment at the fertilization level of 210 kg N ha^−1^. In the second growing season, the greatest values were observed at 140 kg N ha^−1^. Genotype G2 responded with a higher sensitivity to the applied treatments and expressed a positive response to each level of nitrogen application, while the greatest values were observed in the E3 environment at the fertilization rate of 140 kg N ha^−1^. In the second season, the greatest values were observed in environments E6 at the 70 kg N ha^−1^ application level. In the first vegetation season, genotype G3 expressed the greatest values in the E2 environment at the fertilization level of 70 kg N ha^−1^. In the second season, the greatest values were observed in the E6 environment at the 70 kg N ha^−1^ application level. In the first vegetation season, genotype G4 showed the greatest values in the E4 environment at the fertilization level of 210 kg N ha^−1^. In the second season, the greatest values were observed in the E6 environment at the 140 kg N ha^−1^ application level.

In general, average maize yield was lower under all nitrogen conditions in the growing seasons of 2022 compared with the 2021 growing season. The mean yield variation was expected since the maize genotypes showed different responses in different growing conditions. Observed differences in grain yield between the years confirmed the need for multiyear trials to estimate the stability of genotypes, which provided more reliable assessments with smaller margins of error for estimation. Such approaches ensure support for the assessment of genotype adaptation and selection of stable hybrids with high levels of adaptation to different agro-ecological conditions [44,47]. According to Branković-Radojčić [52], a precise assortment of superior genotypes is recommended to increase maize grain yield in certain environments.

The combined ANOVA showed that the phenotypic expression of grain yield of maize was a significant influence on environmental factors, which explained 58.97% of the total variation. The total sum of squares genotype contributed with 5.51%, while interaction displayed 35.5%. In the combined analysis of variance, the main effects, genotypes, and environments were highly significant (35.0 + 374.40)/634.90 and explained 64.5% of the total variation (Table 2). A large sum of squares for the environments indicated that the examined environments were different. Differences among environmental means have caused grain yield variation. Different climatic conditions between the examined growing seasons, as well as various treatments, caused a significant sum of squares for environmental factors in the total variation, which provided high grain yield variation. Lobell et al. [49] reported that the effects of the environmental factors varied between several growing seasons, and environmental factors participated between 50 to 85% of the total treatment variation. Meteorological conditions in the dry 2022 growing season they conditioned for the low average mean yield of all genotypes, while the more productive year was 2022. Such results are in agreement with those obtained by [52,53,54,55].

When measured using environment interaction (GEI), genotype expressed a significant mean square, which indicated that the grain yield of maize genotypes was not stable across different environments. The AMMI analysis showed the complex nature of GEI, which contributed 35.52% of the total sum of squares. The significant GEI in this investigation indicated that grain yield varied between different environments. However, the significant contributions of the main environmental effects and GEI effects in grain yield variation were also reported in previous studies [53,54,55,56,57,58,59,60]. Using the PCA confirmed the statistical significance of the first source of variation (IPCA1), which contributed to the GEI variation with 74.59%. The second main component IPCA2, as the second source of variation, was not expressed as a statistically significant effect in the GEI variation (Table 2). In particular, IPCA1 and IPCA2 contributed to the GEI variation with 74.59% and 21.24%, respectively, and jointly explained more than 95.83% of the GEI variation. Hence, the interaction of maize genotypes within different environments could be best predicted by the first two interaction principal components of genotypes and environments. Such results are logical because the grain yield is a quantitative trait that is controlled by numerous minor genes with great influence over environmental factors. These findings are in accordance with several studies by many authors, who stated that GEI expressed a significant mean square, which suggests that the grain yield of genotypes varied across different environments [54,55].

In the AMMI biplot, the main effects, genotype, and environment for the grain yield of the four maize genotypes are presented on the *x* axis, while the scores of IPCA1 are presented on the *y* axis. The vertical line presents the grand mean for the grain yield, while the horizontal line (y-ordinate) represents the zero of IPCA1 value (Figure 1a). On the basis of the two-dimensional graphical representation of genotypes and environments, it can be denoted that maize genotypes were differing in relation to the main effect, as well as in multivariate effect. According to the biplot (Figure 1a), as well as in terms of the average values (Table 3) based on the E2, E3, and E4 environments points, it can be seen that in these environments in the first growing season of 2021 the maize genotypes establish greater average mean values of grain yield. On the other hand, in the same growing season, it can be observed that in environment E1 (control—without nitrogen fertilization) maize genotypes achieved lower average values of grain yield. On the basis of the distance of the environmental points from the origin it can be also observed that environment E4 (nitrogen application of 210 kg N ha^−1^) had the smallest distance of the environmental points from the origin, which indicates that this environment (E4) can be estimated as the most stable to obtain a stable yield (Figure 1a). Environments E2, E3, and then E1 exposed the greatest interaction values, which indicated that they are less favorable for stable yield production. In the second growing season of 2022, based on the arrangement of the E5, E6, E7, and E8 environments points, it can be seen that the maize genotypes achieved lower mean values of grain yield. On the other hand, in the same growing season, it can be observed that in environment E6 (control—without nitrogen fertilization) maize genotypes achieved the lowest average values of grain yield. Based on the environmental points from the origin, it can also be observed that environment E5 (control) and E6 (nitrogen application of 70 kg ha^−1^) exposed the lowest distance from the origin, which indicates that E5 and E6 can be estimated as the best environments to establish a stable grain yield of maize (Figure 1a).

Environments E7 and E8 show higher and positive interaction values, followed by a higher grain yield of maize in the second growing season. Regarding genotypes dispersion, since it showed small distances from the origin, genotype G3 was found as the most stable in all environments, showing less cross interaction. Genotypes G2, G4, and G1 showed a high distance from the average environment ordinate and exposed unstable reactions in environments. It was observed that genotype G1 was better adapted to the E4 environment (nitrogen application of 210 kg ha^−1^) and showed an average value above the overall mean. Genotype G2 was found to be better adapted to the conditions of the E5 (without nitrogen application) and E6 (nitrogen application of 70 kg ha^−1^) environments and also showed an average value above the overall mean. This result suggests that this genotype has a positive response to the nitrogen application of 70 t ha^−1^ and also showed a stable reaction in conditions in the control environment (E5), which was less favorable. Genotype G4 was found to be well adapted to the conditions of the E7 environment (nitrogen application of 140 kg ha^−1^). According to Gauch and Moran [26], based on a graphical presentation, maize genotypes that are at the top of the graph, such as G1 and G4, showed a positive GEI. On the other hand, maize genotypes that are arranged at the bottom of the graph, such as G2 and G3, showed a negative GEI (Figure 1a).

The presented data are in agreement with our analysis—the AMMI selection of maize genotypes across the investigated environments (Table 3).

Genotype G4 was better adapted to the E1 and E5 environments, of which present control conditions, which means that this maize hybrid showed positive and stable reactions in less favorable conditions without nitrogen fertilizer. In environment E2 (treatment with 70 kg N ha^−1^ application), in the first growing season, genotype G2 was first ranked, implying it had the best adaptability for these soil conditions, followed by genotypes G3, G1, and G4. In the second growing season in environment E6 (treatment with 70 kg N ha^−1^ application), genotype G1 was first ranked, followed by genotypes G4, G3, and G3. This result indicates that these genotypes showed a positive response for soil fertilization treatment with 70 kg N ha^−1^. 

In soil environment E3 (treatment with 140 kg N ha^−1^), in the first vegetation season, genotype G2 was first ranked, indicating that it has great adaptability for these conditions. Genotype G2 was followed by genotypes G3, G1, and G4. In the second vegetation season in soil environment E7 (treatment with 140 kg N ha^−1^), genotype G1 was first ranked, which indicates that this genotype showed a positive response for a soil fertilization treatment with 70 kg N ha^−1^. This genotype was followed by genotypes G4, G3, and G2.

In environment E4 (treatment with 210 kg ha^−1^ nitrogen application), in the first growing season, genotype G2 was ranked first, implying it had the best adaptability for these soil conditions, followed by genotypes G2, G3, and G4. In the second growing season in environment E8 (treatment with 210 kg ha^−1^ nitrogen application), genotype G1 was first ranked, which indicates that this genotype reacted favorably for soil fertilization treatments with 210 kg N ha^−1^.

Overall, it can be concluded that the grain yield of maize had lower values in 2022 compared with 2021 under all the examined conditions. Variation in the mean grain yield is expected since the maize genotypes responded differently in different growing conditions. The high influence of climatic conditions on the grain yield of maize has been reported by [60].

### 2.2. Plant Height

Plant height (PH) is one of the most important traits for grain yield in breeding programs for maize, and is highly heritable despite its complex, polygenic nature [61,62]. On the other hand, certain authors pointed out that plant height, measured over time, can provide an assessment of physiology and critical genetic traits, as well as the influence of environmental conditions on plant performance [63,64,65,66,67,68]. Plant height is associated with the grain number per spike, biomass production, and harvest index, and thus increases grain yield and quality [69].

The mean values of plant height of maize genotypes in the study ranged between 239.4 cm in E4 (nitrogen application of 210 kg N ha^−1^) and 248.8 cm in E2 (nitrogen application of 70 kg ha^−^^1^) in the first growing season. In the second season, the average plant height ranged between 203.2 cm in E5 (control variant with no fertilization applied) and 225.0 cm in E7 (nitrogen application of 70 kg ha^−^^1^) (Table 4). Based on genotypes, the average values of plant height ranged from 225.7 cm (G3) to 236.6 cm (G2). In the first season, at the control variant (E1), the greatest average value for plant height was recorded for maize genotype G1 (270.1 cm), while the lowest mean values of the plant height were observed for genotypes G2 (225.5 cm) and G3 (229.1 cm). In the growing season of 2022, at the control variant (E5), the greatest mean value was observed for genotype G3 (213.5 cm), while the smallest plant height was observed for genotype G1 (184.7 cm). The greatest values of variance for the plant height of maize were in the E2 environment, while the lowest values were observed in the E8 environment. The standard deviation for the plant height varied from 1.35 in the E8 environment to 22.47 in the E2 environment (Table 4). According to the AMMI stability value (ASV ranking) for the plant height, the maize genotype G4 had the lowest value, which showed that it was the most stable, while genotype G2 had the greatest value and the lowest stability (Table 4).

The results of the present investigation revealed that different environmental conditions, as well as the growing season, caused significant plant height variation between genotypes. However, apart from the different mean values that were observed, in general, a higher sensitivity of plant height and the lowest mean values were observed in the second vegetation season due to the meteorological conditions. Favorable climatic conditions in the first growing season slightly reduced the differences between the average values of plant height in different environments. In the second growing season, a higher sensitivity for plant height was observed, which was mostly caused by drought conditions and less rainfall in the first part of summer, especially during June and July. Therefore, as a consequence, the effects of fertilization measures were the most obvious. The plant height of certain maize genotypes was constant in various environments, while certain maize genotypes showed significant variation across different treatments. As we observed, hybrids expressed higher plant height in the first and more favorable growing season of 2021 than in the growing season of 2022. This means that the hybrids were less stressed by the June and July temperature peaks when they progressed through silking and pollination stages at a faster rate. This result is expected since the plant height of maize is a quantitative and variable trait that highly depends on environmental factors. These findings were similar to those of [62].

Based on the ANOVA, all sources of total variation were found to be statistically significant (environmental and genotypic main effect), as well as agronomically important. The ANOVA results revealed that the plant height expression was highly significantly affected by environmental factors, since that significant variance explained 53.32% (at the 1% level) of the total variation, while genotype participated with 4.73% of the total variation in the trial (Table 5).

Environments showed a large sum of squares, which pointed out that examined environments were diverse and caused variation in the plant height. A non-additive component, GEI, was highly significant, contributing with 41.95% of the total variation. Large differences among environments and vegetation seasons caused a high sum of squares of environmental factors in the overall variation in the trial, which indicated that they are the most responsible for variations in the plant height of maize.

This result is expected since during the trial period of two growing season, significant differences were noticed regarding sums and amounts of precipitation, as well as average temperatures in critical phases of maize development, such as tasseling, silking, pollination, and fertilization, as well as in the grain filling stages (periods from June to August). Such results are in agreement with results reported by Branković-Radojčić et al. [52]. Such results are logical because the plant height of maize is a quantitative and variable trait and influenced by environmental conditions [16,21]. Using PCA, we discovered the statistical significance of two principal components, IPCA 1 and IPCA 2, while the residual accounted for only 3.8% of the total variation (Table 5). IPCA1 participated in the GEI variation with 52.53%, and IPCA2 participated in the GEI variation with 38.40%. It was noticed that both principal components had statistically significant effects on the GEI and explained 90.93% of the variation in the GEI (Table 5).

Based on biplot and the graphical representation of genotypes and environments, it was observed that maize genotypes differed in relation to their main effect, as well as in their multivariate effect. Based on the points of E1, E2, E3, and E4 on the biplot, it can be noticed that in these environments, which belong to the first growing season of 2021, the maize genotypes had the greater values of grain yield. On the other hand, during the same growing season it can be observed that in environment E1 (control—without nitrogen fertilization) maize genotypes achieved lower average values of plant height than in environments E2 and E3. Environment E1 (control—without N application) had the largest distance from the origin, which revealed that this environment cannot be used as a stable environment for stable plant height formation. Environments E4, E3, and E2 showed lower interaction values, which indicated a stable response, achieving stable maize plant height (Figure 1b).

In the second growing season of 2022, based on the position of the environment points for E5, E6, E7, and E8, it can be concluded that the maize genotypes showed lower mean values for plant height. On the other hand, during the same growing season, it can be observed that in environment E5 (control—without nitrogen fertilization) maize genotypes achieved the lowest values of plant height. Based on the position of the environmental points from the origin it can be observed that environment E6 (nitrogen application of 70 kg ha^−1^) and E8 (nitrogen application of 140 kg ha^−1^) showed the smallest distances for the environmental points from the origin to zero point. This revealed that environments E6 and E8 can be estimated as the most stable environments for the maize plant height. Environment E7 showed lower stability, while environment E5 showed higher and negative interaction values, followed by lower maize plant height in the second growing season (Figure 1b). Regarding genotype dispersion, since they showed small distances from the origin, maize genotypes G3 and G4 expressed the most stable reactions across the environments and also showed less cross interaction. Genotypes G1 and G2 had a large distance from the ordinate and expressed a less stable response in their environments. Genotype G1 was stable and with good adaptability to the conditions of environment E1 (without nitrogen application) and showed a higher mean value than the grand mean. Genotype G2 was better adapted to the conditions of the E2 (nitrogen application of 70 kg ha^−1^) and E3 (nitrogen application of 140 kg ha^−1^) environments and also showed an average value above the overall mean. This indicates that this genotype has a positive response to the nitrogen treatment in amounts between 70 kg ha^−1^ and 140 kg ha^−1^. Genotype G3 was better adapted within the conditions of the E6 (nitrogen application of 70 kg ha^−1^) and E8 environments (nitrogen application of 210 kg ha^−1^). Genotype G4 was better adapted to the conditions of the E7 environment (nitrogen application of 140 kg ha^−1^). As it was pointed out, genotypes with a position at the top of the graph, such as genotypes G1 and G4, showed a positive GEI, while genotypes G2 and G3, which were at the bottom, showed negative GEI (Figure 1b). 

AMMI selections per environment indicated that genotype G1 is the most promising genotype for environments E1 (control—soil without nitrogen application in the growing season of 2021), E4 (soil with nitrogen application in amount of 210 kg ha^−1^ in the growing season of 2021), and E7 (soil with nitrogen application using granular urea in the amount of 140 kg ha^−1^ in the growing season of 2022). Genotype G2 was better adapted to environments E2 (soil with nitrogen application using granular urea in the amount of 70 kg ha^−1^ in the growing season of 2021), E3 (soil with nitrogen application using granular urea in the amount of 140 kg ha^−1^ in the growing season of 2021), and E8 (soil with nitrogen application using granular urea in the amount of 200 kg ha^−1^ in the growing season of 2022). Genotype G3 was best performing in the E5 environment (control—soil without nitrogen application in the growing season of 2022). Genotype G4 was better adapted to the E6 environment (soil with nitrogen application using granular urea in the amount of 70 kg ha^−1^ in the growing season of 2022). Genotype G6 was ranked within the first four in all environments and showed consistency, implying it had the best adaptability for all soil conditions (Table 6).

Maize genotypes which maintain their plant height in control environments are well adapted in limited soil conditions without fertilization, since that expression of plant height presents a high capacity of stems to accumulate sufficient stem reserves for the partitioning to grain [48].

### 2.3. Kernel Number per Row

Kernel number per row (KRN) is an important yield trait in maize and directly affects grain yield [65]. Since kernel number is strongly affected by environmental stresses, the potential ear length varies from year to year depending on growing conditions. Traits of maize yield, such as kernels number per row, as well as kernels per ear and rows per ear, are highly related with grain yield and stress resistance. Therefore, these traits can be used as quantitative indices for selecting superior maize varieties and analyzing trial results in maize breeding [66].

The average values of the kernel number per row in the study in the first growing season ranged within different environments, between 28.81 in the E1 environment (control) and 37.17 in the E3 environment (at the nitrogen application of 140 kg N ha^−1^). In the second season, the average values of the kernel number per row ranged within different environments from 27.36 (E5) to 34.47 in the E7 environment with a nitrogen application of 140 kg N ha^−1^. The genotype average values of the kernel number per row ranged from 30.62 (G3) to 36.89 (G1) within different genotypes. Regarding the environment mean for both seasons, the greatest values were observed within the E3 and E7 environments, both for the nitrogen application of 140 kg N ha^−1^. In these environments, the greatest values were observed for genotype G1, which had values of 41.36 and 40.98 in the second vegetation season (Table 7).

For the control variant (E1) without fertilization, the highest average value for the kernel number per row was recorded for maize genotype G1, having a value of 30.90 in the first growing season, while the lowest mean value was noted for genotype G2 (23.63). In the growing season of 2022 for the control variant (E5), the highest average value for the kernel number per row was noticed for genotype G3, having a value of 29.82, while the lowest mean value was denoted for genotype G2 (23.63). 

With the nitrogen application conditions of 70 kg ha^−1^ (E2), the highest average value for the kernel number per row was denoted for the maize variety G1, having a value of 37.29 in the first growing season, while the average value was observed for genotype G2 (33.93). In the second growing season, for the nitrogen application conditions of 70 kg N ha^−1^ (E6), the highest value for the kernel number per row was estimated for maize variety G3, having a value of 36.60, while the lowest mean value was denoted for genotype G2 (32.30). The greatest values of variance for the kernel number per row were observed in the first growing season in the control of the E1 environment, while in the second growing season, the observed values of variance were the highest in the E8 environment (for the nitrogen application of 210 kg N ha^−1^) (Table 7). The standard deviation for the plant height varied from 1.64 in the E2 environment to 4.33 in the E4 environment (Table 7). According to the ASV ranking for the kernel number per row of maize, maize genotype G4 had the lowest value, which showed that it was the most stable, while genotype G1 with the greatest values had the lowest stability (Table 7).

In nitrogen application conditions of 140 kg N ha^−1^ (E3), the highest average value for the kernel number per row was estimated for maize variety G1, having a value of 41.36 in the first growing season, while the lowest mean value was denoted for genotype G4 (33.59). In the second growing season with the nitrogen application conditions of 140 kg N ha^−1^ (E7), the highest average value for the kernel number per row was recorded for maize variety G1, having a value of 40.98, while the lowest mean value was denoted for genotype G3 (31.19) (Table 7).

Maize genotype G1 had the highest value for the kernel number per row of 41.07, which was with the nitrogen application condition of 210 kg N ha^−1^ (E4) in the first growing season, while the smallest average value was observed for genotype G2 (31.79). In the second growing season with nitrogen application conditions of 210 kg N ha^−1^ (E8), the highest average value for the kernel number per row was observed for maize variety G1, having a value of 38.98, while the lowest mean value was observed for genotype G3 (25.49) (Table 7).

The presented results showed that significant variations in the kernel number per row of maize were noticed due to different environment conditions and levels of fertilization, but this was also dependent on maize genotype used in the study, as well as on growing season. It was observed that there was a higher sensitivity for the kernel number per row of maize trait in the second growing season because the average values were the lowest in the control variant, as well as with another treatment. In the second season, a higher sensitivity of this trait was mostly caused by drought conditions during tasseling, silking, and pollination. Less precipitation in this part of the season greatly reduces the potential kernel number per row. More appropriate growing conditions in the first season encourage a high potential kernel number. This result is expected since the kernel number per row of maize is an important, quantitative trait whose expression largely depends on environmental factors. This is in agreement with the high values of variation with values until 32.02% (Table 7).

In general, the growing season of 2021 provided generally higher mean values of the kernel number per row than the growing season of 2022. Maize genotypes responded differently to fertilization measures, enhancing the kernel number in more favorable agro-ecological conditions. The effects of fertilization measures and variation in the kernel number per row in maize were influenced by climatic conditions. The grain number of maize is a trait that is affected by assimilates accumulation during the critical period of kernel number determination [67]. According to Božović et al. [3], in drought conditions, due to the lack of water in the grain filling stage, the main yield traits of maize are the number of kernels, seed weight, and seed filling rate. The presence of stress during pollination has a high influence on the duration of the pollination and silking phase, affecting the number of kernels per row, the number of kernels per ear, and other yield components [67]. Climatic factors have a great influence on yield [68,69,70,71,72,73,74,75].

The AMMI analysis of variance revealed significant effects, both for maize genotypes and environments (Table 8).

The AMMI analysis showed a highly significant influence for all three sources of variation, genotypes, treatments, and their interactions, and had significant effects on the phenotypic variation in the trait kernel number per row of maize (Table 8).

This means that differences in kernel number per row among genotypes is due to genetic as well as environmental factors. In the analysis of variance, the genotypes and environments as the main effects were highly significant (591.5 + 1015)/2030.1, explaining 79.14% of the total variation. The combined ANOVA showed that the kernel number per row of maize was significantly influenced by the environment, since that significant variance at the 1% level explained 50.0% of the total variation, while the genotype contributed with 8.22% of the total variation in the trial. A large sum of squares for environments revealed that the trial environments were diverse, with large differences among environmental means, causing variation in the kernel number per row of maize. Genotype by environment interaction showed a significant mean square, which indicated that the kernel number per row was variable within different environments. The AMMI model showed the complex nature of the GEI, which presented 20.87% of the total sum of squares of the trial. The PCA, as an additional analysis of the GEI, pointed out the statistical significance of the two main principal components (IPCA 1 and IPCA 2), while the residual accounted for only 1.4% of the total sum of squares (Table 8). IPCA1 participated in the GEI variation with 56.26%, while IPCA2 participated in the GEI variation with 36.92%, both with statistically significant effects on the GEI variation. These two main principal components together explained about 93% of the variation in the GEI. This result is expected since the kernel number per row of maize is a quantitative trait under the high influence of environmental factors (Table 8).

The AMMI biplot analysis indicated that environment E6 exposed the lowest distance of the environmental points from the zero point, which led to the conclusion that environment E6 was the most stable environment for the kernel number per row of maize (Figure 2). Environmental points E2, E3, E4, E6, and E7 also had a small distance from the origin, which indicated that these environments were stable for the plant height trait. In these environments, the maize genotypes achieved higher average values of kernel number per row. According to the arrangement of the E1 and E5 points, which correspond to control variants in both seasons, and the E8 environment, maize genotypes achieved lower values than the average mean of the kernel number per row of maize. Apart from the lowest values which were observed, these environments showed the greatest interactions and largely contributed to the GEI. High interaction values characterized these environments as unfavorable for the kernel number per row of maize trait.

Regarding the genotypes position on the graph, it can be seen that genotype G2 expressed the most stable reaction for the observed trait, showing a PCA1 score close to zero across all environments, indicating almost no cross interaction. Genotypes G1 and G3 were far away from the environmental ordinate, indicating that these genotypes were unstable across the environments. Genotype G3 was well adapted to the unfavorable conditions of the control environments of E1 and E5, having its average below the overall mean. Genotype G3 showed high interaction values, while genotype G4, with its average within the overall mean, was better adapted to the conditions of the E2 and E6 environments, which corresponded to the level fertilization of 70 kg N ha^−1^ nitrogen applied in both seasons. Maize genotype G1 showed a positive response to fertilization measures since it has a greater average value of number of kernels per row in environments E3 and E7, which corresponded to the level of fertilization of 140 kg N ha^−1^ nitrogen. However, genotypes with a position at the top of the graph, such as G2 and G1, showed a positive GEI. Genotypes which have a position at the bottom of graph, such as genotypes G3 and G4, showed a negative GEI [26] (Figure 2). 

Based on the AMMI selection of maize genotypes across the examined environments, it can be seen that the genotype G1 was ranked as the first, and it can be estimated as the genotype with the best stability for the different conditions of nitrogen supply (Table 9). Genotype G4 was found to be better adapted to the stressful conditions of the E5 environment, which corresponded to control variants, in the second growing season. In the control environment (E1), without fertilization, genotype G1 showed the best adaptability, since it ranked first. In the E5 environment (control; soil without fertilization) in the growing season of 2022, genotype G4 was estimated as the most promising and well adapted. Such results mean that these maize genotypes (G1 and G4) have good stability and positive response in the conditions of the control environments without nitrogen fertilizer.

The results of the first AMMI selection per environment showed that a greater sensitivity of the kernels number per row was observed in less favorable conditions in the second growing season (Table 9).

The mean of the kernel number per row was smaller in 2022 than in the 2021 season under all conditions. The observed average mean variation indicated that the maize genotypes reacted differently in different growing conditions. In the second season, the overall average value of the number of kernels per row was the smallest in the control variant without fertilization measures. Furthermore, smaller mean values were observed in the second season for each of the studied treatments. This result is expected, since the number of kernels per row is a variable trait that largely depends on the climatic conditions throughout the season. The results indicated that the kernel number per row trait is important for the yield production of maize. This trait is under high influence of climatic conditions, water shortages, and high temperatures during the critical growth stages. The responses of maize plants to water and nitrogen stress may cause further effects beyond the individual impacts [75,76]. It can also be observed that the high yield of different maize hybrids is simultaneously related to different yield components. This leads to the conclusion that the high values of one of the yield components do not necessarily mean high yield potential. Yield is a complex trait; hence, the genotype that has a balanced ratio and high values of certain yield components will result in a higher yield. These results are in agreement with those obtained by [77]. However, the obtained results indicated that in maize genotypes, aside from the effects of different treatments, climatic conditions were affected by significant differences in certain yield traits. Similar findings regarding the importance of climatic factors have also been reported by [77,78,79,80].

Therefore, the observed different responses of genotypes within various environments indicated the need for the evaluation of various maize genotypes in order to make optimal recommendations and selections of the optimal genotype for the specific growing conditions.

Higher yields and biomass production of the hybrids with the latest genetic stock can be associated with increased plant nutrient uptake and higher utility of soil nutrient content [81]. Optimal nitrogen supply has a significant role in the growth characteristics of plants, as it is the main factor for plant cell components, primarily due to its role played in the photosynthetic apparatus [82]. The nitrogen use efficiency (NUE) of maize is estimated at 33% globally, which is negatively affected by fertilizer leaching under the root zone and denitrification [83]. According to Surendran et al. [84], maize absorbs about 10–20% of its total nitrogen requirements up to the V4 stage, whereas during 6 weeks of growth from V4 to VT, N accumulation approaches 60–70% of total N uptake. Growth stages have different nutrient demands based on their actual demand. Nitrogen and potassium had their maximum effect on the stalk of maize during the growing season. Magnesium and copper were the second most important and desirable factors during the different growth stages and treatments in relation to the stalk. Nitrogen and calcium had their maximum impact during the yield formation stage, and nitrogen and phosphorus had their most desirable effect during the grain filling stage [85]. Total nitrogen was an essential factor of maize in all growth stages. Increased nitrogen fertilizer leads to more dry matter production and grain yield. Increasing nitrogen accelerates green growth, increases the above-ground mass of the plant, increases the evaporation of plants, and causes the roots to expand and bulk up [86]. Numerous reports recorded the positive effect of nitrogen on grain growth per grain, grain weight, and grain yield of maize, with a tendency to use higher amounts of nitrogen fertilizer [87,88,89,90,91].

## 3. Materials and Methods

### 3.1. Field Trial

The present study was carried out at the experimental trial field using chernozem-type soil on location in Ravno Selo (45.441° N, 19.6321° E), in Bačka region, Vojvodina Province (Serbia), during two consecutive growing seasons in 2021 and 2022. The experimental material in the study was comprised of four maize genotypes (*Zea mays* L.) of different maturity classes and lengths of vegetation period. Therefore, four varieties of maize, namely, P0725, which belongs to FAO 600 (G1); P9889, which belongs to FAO 410 (G2); P9757, which belongs to FAO 340 (G3); and P9074, which belongs to FAO 230 (G4), were selected, and four different levels of nitrogen were applied in the experiment in order to assess the variability in the yield-related traits. The maize cultivars used in the study were released by Pioneer Hi-Bred International, Inc., producer of seeds for agriculture. All maize genotypes were suitable for maize production in agro-ecological conditions of Serbia. The field trial was sown at 70 cm inter-row spacing and 20 cm spacing between plants in the row. Planting in both growing seasons was completed by the middle of April, while harvest ended in the first week of October. The experimental trial was set up on the chernozem type of soil according to a completely randomized block design (RBCD), with four different nitrogen treatments and three replications of each treatment. The study included four different N treatments in amounts of 0, 70, 140, and 200 kg N ha^−1^. The application of nitrogen mineral fertilizers (N) was applied pre-planting by incorporating granular urea (46% N) after agrochemical analysis of soil, while weed and pest control on field was performed by using standard cultivation practice. Each plot contained 4 rows of maize 14 m long, and the central part of the 2 middle rows was measured, while the side rows were not considered as they served as guard rows. In the field trial, each treatment in one growing season was considered as a special environment, which produced eight different agro-ecological environments, equal in agro-technical terms but in different nitrogen application treatments of soil. The eight analyzed environments were labeled as follows: E1 represents control, soil without nitrogen application (control) in the growing season of 2021; E2 represents soil with nitrogen application using granular urea in the amount of 70 kg N ha^−1^ in the growing season of 2021; E3 represents soil with nitrogen application using granular urea in the amount of 140 kg ha^−1^ in growing season of 2021; E4 represents soil with nitrogen application using granular urea in the amount of 200 kg ha^−1^ in the growing season of 2021; E5 represents control, soil without nitrogen application (control) in the growing season of 2022; E6 represents soil with nitrogen application using granular urea in the amount of 70 kg ha^−1^ in the growing season of 2022; E7 represents soil with nitrogen application using granular urea in the amount of 140 kg ha^−1^ in the growing season of 2022; and E8 represents soil with nitrogen application using granular urea in the amount of 210 kg ha^−1^ in the growing season of 2022 (Table 10).

At the stage of full maturity, the central part of the two middle rows was measured, while the side rows were not considered and served as guard rows. At the stage of full maturity, ten average maize plants from each replication of each plot were selected separately, and yield traits, such as a plant height (cm), grain number per row, and (%) number rows per ear, were estimated (Table 10).

Plant height (PH) measurements were taken at mid anthesis as the distance from the ground surface to the node bearing the flag leaf. Maize grain yield (GY) was only measured for the net plot area as the two border plants close to the alley were discarded. GY was obtained by harvesting each maize cultivar plot separately, and was adjusted to the 14% moisture content and expressed in t ha^−1^.

### 3.2. Soil Conditions

The field trial was set up using chernozem-type soil in Bačka (Vojvodina) Province, Serbia. This chernozem-type soil is characterized by favorable physical and chemical properties, stable aggregates, good crumbly structure, and good water permeability. This type of soil is suitable for agricultural production since it is well provided with humus, mineral nitrogen, and plant nutrients. The chernozem soil properties until a depth of 30 cm, from which soil samples were taken to establish the experiment, are presented in Table 11.

### 3.3. Meteorological Conditions

During investigation, meteorological data, temperature, and precipitation were very variable. Crop production is highly sensitive to climate, while climate change significantly affects crop production [35,36,37]. During the investigation, the temperatures, amounts, and schedule of precipitation during both maize vegetation periods were analyzed. The meteorological conditions, monthly precipitation, and air temperatures for 2021 and 2022 during the trial were taken from the hydro-meteorological service of the Republic of Serbia, situated in Zmajevo, near to the field trials (Figure 3).

Climatic conditions in the second vegetation season were significantly more variable and less favorable compared to the conditions in the previous season. The weather conditions in the first vegetation season of 2021 were more favorable for maize production than climatic conditions of the 2022 growing season. Climatic conditions in the second vegetation season were significantly more variable compared to the conditions in the previous season. In the second half of May, June, and July, the average daily air temperatures significantly exceeded the multi-year average value. They were characterized by extreme heat. The critical maize development phases (tasseling, silking, and pollination), including the grain formation and development, took place in extreme drought conditions. Air temperature values were generally higher than optimal at the time of grain filling, while soil moisture reserves were rapidly depleted due to intensive evapotranspiration [32]. Average temperatures in the growing season in 2022 were 19.46 °C and were higher by 0.98 °C compared to growing season of 2021, while total precipitation in 2022 was higher by 31.6 mm, but with an unfavorable schedule, especially in critical stages of maize. According to the percentile method, the average summer air temperature in the entire territory of Serbia was categorized as extremely warm (Figure 3). Precipitation and temperature have a decisive influence on the yield [90,91,92,93,94,95,96].

### 3.4. Statistical Analyses

In the present study, the genotype–environment interaction (GEI) was evaluated using the additive main effects and multiplicative interaction model (AMMI model) developed by [38]. This model integrates standard analysis of variance (ANOVA) and principal components analysis (PCA) into a single statistical model [38].

Although GEI possesses agronomical and genetically important effects, and the sum of squares accounts for a large proportion of the total variation, the ANOVA test of GEI is frequently not significant due to the high degree of freedom [39]. The AMMI model was used to separate the genotype main effect, environment main effect, and GEI, while additional GEI analysis can be carried out using principal component analysis (PCA). Therefore, the model can be summarized by the following equation:Y_ger_ = μ + α_g_ + β_e_ + Σλ_n_ξ_gn_η_en_ + Θ_ge_ + ε_ger_, 
where y_ge_ presents the mean of maize yield or other observed trait for genotype g in the environment e, μ—grand mean, α_g_—genotypic mean deviations, β_e_—environmental mean deviations, n—number of PCA axis retained in the adjusted model, λ_n_—eigenvalue of the PCA axis n, ξ_gn_—genotype score for PCA axis n, η_en_—score eigen vector for PCA axis n, Θ_ge_—residual, and ε_ger_—experimental error.

This model has a graphical interpretation that can estimate yield performance and stability simultaneously, as well as discover optimal testing environments [40]. The model uses graphs, called biplots, as a graphic presentation for both the main and interaction effects for genotypes and environments concurrently, presenting in more detail an analysis of the GEI [20,27,41]. In the AMMI analysis, the interaction principal component (IPCA1) score of a genotype was used as an indicator of the stability of a genotype over environments. Zero IPCA value points out the genotypes with highest stability. The IPCA value with long distance from zero indicates genotype instability [35,42]. When a genotype and environment have the same sign on the PCA axis, their interaction is positive, and if it is a different sign, their interaction is negative. On the biplot, genotypes and environments with high PCA1 scores, either positive or negative, have high interactions, while those with PCA1 scores of zero or near to zero have a small interaction [30,40]. The data analysis was performed using the program GenStat 9^th^ Edition statistical software package (trial version), VSN International Ltd. [43].

In addition to the additive main effects and multiplicative interaction model (AMMI model) developed by [38], we additionally provided for a quantitative stability measure in order to quantify and rank genotypes by their yield stability according to the Purchase JL [91], which is described as:$$\mathrm{ASV}=\sqrt{(\frac{\mathrm{IPCA}1\;\mathrm{SS}}{\mathrm{IPCA}2\;\mathrm{SS}}\times {\left(\mathrm{IPCA}1\;\mathrm{score}\right)}^{2})+{\left(\mathrm{IPCA}2\;\mathrm{score}\right)}^{2}}$$
where ASV = AMMI stability value, SS = sum of squares, IPCA1 = interaction of principal component analysis one, and IPCA2 = interaction of principal component analysis two, while IPCA1 SS/IPCA2 SS squares present the weight derived from dividing the sum of IPCA1 SS/IPCA2 SS squares by the sum of IPCA2 squares. The larger the absolute value of the IPCA mean, the greater the adaptability of a certain genotype for a specific environment, while lower ASV values indicate larger stability in different environments.

### 3.5. Standard Deviation

Variance components were calculated by standard deviation. Standard deviations (S) were taken as indicators of the variability of searched characteristics, calculated by the equations indicated.
$$S=\sqrt{\frac{{\sum \left(x-\bar{x}\right)}^{2}}{N-1}}$$

Standard deviation was calculated to test the significance of differences between mean values of the characteristics. The standard deviation is the average amount of variability in the dataset. It tells, on average, how far each value lies from the mean. A high standard deviation means that values are generally far from the mean, while a low standard deviation indicates that values are clustered close to the mean [92,93,94,95,96]. 

## 4. Conclusions

An important issue when using fertilizers is the comparison of methods and amounts of fertilizer used, which is very important from the aspect of increasing maize yield and economic production. The results of the investigation indicated that the grain yield, plant height, and number of kernels per row of examined maize genotypes were highly influenced by GEI effects. Additive main effects and multiplicative interaction (AMMI) analysis showed a significant effect on the genotypes, environments, and the GEI for all traits. The significance of the GEI effect indicated that maize genotypes responded differently to various environmental conditions. Varying fertilizer levels create environments that hybrids respond differently to because of the differences in their genotypic constitution. The weather showed that it is one of the most important factors in maize production. A long period without rain and a deficit in rain amounts, combined with extremely high air temperatures, especially in the generative phase, resulted in lower yields in the season of 2022. Changing weather conditions considerably influenced the effect of nitrogen fertilizations on the observed variation in the grain yield traits of maize. The first growing season under investigation was followed with more favorable weather conditions for maize crop development, namely, increased precipitation, while the second growing season was dry, with an unfavorable schedule and a lower amount of precipitation, especially during the critical stage of maize growth, which affected average yields more than variations in genotypes and treatments. Th more favorable weather conditions of the first growing season exhibited a significant impact on achieving the higher mean values of all observed traits and reduced the differences between the mean values per treatment. 

For all of the studied traits, the AMMI analysis and ASV ranking were in accordance and showed similar results. Based on the tested genotypes, genotype G3, besides a lower mean value than average, expressed the most stable reaction over all environments for the grain yield of maize. In this analysis, maize genotype G1 expresses the greatest value of grain yield, but also shows high interaction values, which indicates it as the least suitable for the stable establishment of grain yield. Regarding the plant height trait, genotype G3, besides a lower mean value than the average mean of this trait, expressed the most stable reaction over all environments. Regarding the kernel number per row of maize trait, genotype G2, with a lower mean value than average, expressed the most stable reaction over all environments. 

This research revealed that the plant height, number of kernels per row, and grain yield of the maize were important factors when optimizing fertilizer use on the basis of sustainable agriculture with maize hybrids. The study showed variation for all studied traits among the maize genotypes, which indicates sufficient variability that can be exploited through selection, as well as for the recommendation of suitable maize genotypes in the examined environments. Information on the effect of fertilizers on maize yield components could assist in maintaining the balance and stability of grain yield when using application fertilizers. The stable maize hybrids could be recommended for both N stress and optimum nitrogen conditions. The maize hybrids identified for the various N conditions will support seed companies to make decisions on the maize varieties to produce for agricultural production. Therefore, the constraints to maize production due to soil fertility can be reduced. The effect of nitrogen on the quantitative properties of maize showed that nitrogen increases the production of grain yield and its components. The determination of the fertilization effect on the grain yield and partitioning of modern maize hybrids analyzed with different multivariate statistical methods can provide a basis for the farmers to implement hybrid and site-specific nutrient management systems with the goal of reducing the environmental impact of over fertilization, ensuring more rational production.

## Figures and Tables

**Figure 1 plants-12-02165-f001:**
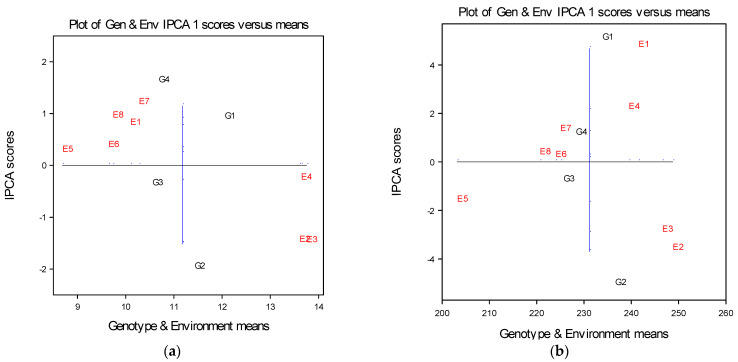
(**a**) AMMI 1 biplot of four maize genotypes across eight environments (two years × four treatments) for the estimation of main and multivariate (GEI) effects for the grain yield in tha^−1^ at 14% moisture; (**b**) AMMI 1 biplot of four maize genotypes across eight environments (two years × four treatments) for the estimation of main and multivariate (GEI) effects for the plant height of maize.

**Figure 2 plants-12-02165-f002:**
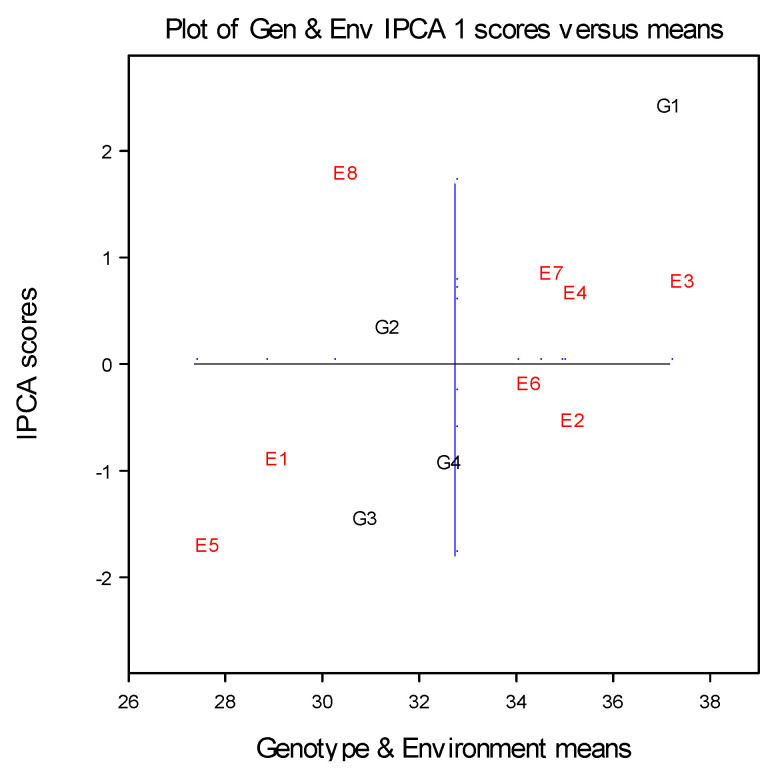
AMMI 1 biplot of four maize genotypes grown across eight environments (two years × four treatments) for the estimation of main and multivariate (GEI) effects for the number kernels per row.

**Figure 3 plants-12-02165-f003:**
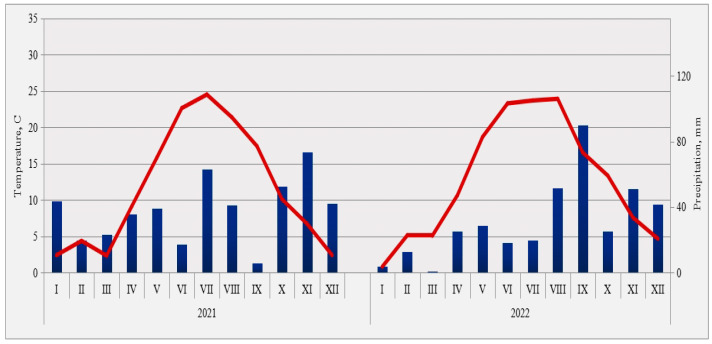
Temperature (°C, red line) on left axis and precipitation (mm, pillars) on the right axis during 2021–2022, Serbia.

**Table 1 plants-12-02165-t001:** Average values for grain yield of maize (t/ha) and interaction principal component values of IPCA1 and IPCA2 of the AMMI model, and variance, standard deviation, and AMMI stability value (ASV) of four maize genotypes grown in eight different environments.

Parameter	Genotypes									
Environment	G1	G2	G3	G4	Em	IPCAe *	IPCAe *	Mean	Variance	SD
E1	10.86	8.69	9.68	11.15	10.10	0.74688	0.50283	10.10	1.28	1.13
E2	12.01	16.69	14.39	11.31	13.60	−1.50830	0.76086	13.60	5.98	2.45
E3	14.53	17.30	12.91	10.27	13.75	−1.51677	−0.81461	13.75	8.68	2.95
E4	15.32	14.71	12.39	12.13	13.64	−0.30621	−0.71735	13.64	2.60	1.60
E5	7.95	8.13	9.21	9.45	8.68	0.22876	1.19637	8.68	0.56	0.75
E6	10.68	9.22	8.96	9.71	9.64	0.32010	0.07353	9.64	0.58	0.76
E7	12.67	8.30	8.81	11.32	10.27	1.14637	−0.34905	10.27	4.29	2.07
E8	12.36	8.36	8.05	10.15	9.73	0.88918	−0.65259	9.73	3.92	1.98
IPCAg [*]	0.8708	−2.024	−0.416	1.5686	11.18					
IPCAg [*]	−1.507	−0.282	1.0408	0.7485						
Gm	12.05	11.43	10.55	10.68						
ASV	3.4089	7.1116	1.7928	5.5588						

Gm: Genotype average mean; Em: Environmental average means; IPCAe1: The first-interaction principal component axes for environments; IPCAg1: Interaction principal component axes for genotypes; IPCAe2: The second-interaction principal component axes for environment; IPCAg2: The second-interaction principal component axes for genotype; G: Maize genotypes; E: Environmental labels with control (K) and 70, 140, and 210 kg ha^−1^ nitrogen applied in both maize growing seasons; V: Variance; SD: Standard deviation; ASV: AMMI stability value.

**Table 2 plants-12-02165-t002:** AMMI ANOVA for the grain yield of four maize genotypes across eight environments.

Source ^1^	Df	SS	MS	F-Value	F_Prob	The Share of Total Variation %
Total	95	999.70	10.52	-	-	-
Treatments	31	634.90	20.48	3.76 **	0.0000	63.51
Genotypes	3	35.00	11.67	2.14 ^ns^	0.1074	5.51
Environments	7	374.40	53.49	8.30 **	0.0000	58.97
Block	16	103.10	6.44	1.18 ^ns^	0.3156	16.24
Interactions	21	225.50	10.74	1.97 *	0.0266	35.52
IPCA 1	9	168.20	18.68	3.43 **	0.0025	74.59
IPCA 2	7	47.90	6.84	1.25 ^ns^	0.2929	21.24
Residuals	5	9.50	1.89	0.35 ^ns^	0.8815	1.50
Error	48	261.70	5.45	-	-	-

^1^ **: Highly significant at *p* < 0.01 probability level; *: Significant at *p* < 0.05 probability level; ^ns^: Not significant; Df: Degree of freedom; F: F value calculated; IPCA1: The first-interaction principal components axes; IPCA2: The second-interaction principal components axes.

**Table 3 plants-12-02165-t003:** The first AMMI selections per environment.

No.	E	Mean	IPCA	Genotypic Rank			
		(tha^−1^)	Score	1	2	3	4
7	E7	10.27	1.1464	G1	G4	G3	G2
8	E8	9.73	0.8892	G1	G4	G2	G3
1	E1	10.10	0.7469	G4	G1	G3	G2
6	E6	9.64	0.3201	G1	G4	G2	G3
5	E5	8.68	0.2288	G4	G3	G2	G1
4	E4	13.64	−0.3062	G1	G2	G3	G4
2	E2	13.6	−1.5083	G2	G3	G1	G4
3	E3	13.75	−1.5168	G2	G1	G3	G4

E: Environments; IPCA score: Score based on the values of the first interaction principal component axes.

**Table 4 plants-12-02165-t004:** Average values for the plant height of maize (cm) and interaction principal component values of IPCA1 and IPCA2 of the AMMI model; the variance, standard deviation, and AMMI stability value (ASV) of four maize genotypes grown in eight different environments.

Parameter	Genotypes									
Environment	G1	G2	G3	G4	Em	IPCAe [*]	IPCAe [*]	Mean	Variance	SD
E1	270.1	225.5	229.1	241.4	241.5	4.6714	0.7512	241.50	409.2	20.23
E2	242.1	282.1	236.4	234.4	248.8	−3.6979	2.6553	248.80	505.0	22.47
E3	240.8	273.3	236.6	235.3	246.5	−2.9431	1.8387	246.50	324.7	18.02
E4	263.0	244.2	220.5	230.0	239.4	2.1095	3.0493	239.40	341.9	18.49
E5	184.7	205.0	213.5	209.4	203.2	−1.7123	−3.7575	203.20	163.3	12.78
E6	216.8	218.4	230.4	230.4	224.0	0.1285	−3.1318	224.00	55.0	7.42
E7	232.3	222.9	220.1	224.7	225.0	1.2045	−0.4208	225.00	27.3	5.22
E8	221.3	221.7	218.7	221.0	220.7	0.2395	−0.9845	220.70	1.8	1.35
IPCAg [*]	4.9738	5.1447	0.8747	1.0456						
IPCAg [*]	3.3780	3.3260	3.8116	2.8924						
Gm	233.9	236.6	225.7	228.3						
ASV	7.5969	7.7847	3.9950	3.2267						

Gm: Genotype mean; Em: Environmental mean; IPCAe1: The first-interaction principal component axes for environment; IPCAg1: The first-interaction principal component axes for genotype; IPCAe2: The second-interaction principal component axes for environment; IPCAg2: The second-interaction principal component axes for genotype; G: maize genotypes; E: Environmental labels with control (K) and 70, 140, and 210 tha^−1^ nitrogen applied in both maize growing seasons; V: Variance; SD: Standard deviation; ASV:AMMI stability value.

**Table 5 plants-12-02165-t005:** AMMI ANOVA for the plant height of four maize genotypes studied across eight environments.

Source	df	SS	MS	F-Value	F_Prob	The Share of Total Variation %
Total	95	52,208	549.6	*	*	
Treatments	31	38,335	1236.6	7.03 **	0.000	73.43
Genotypes	3	1812	603.9	3.43 *	0.024	4.73
Environments	7	20,441	2920.2	8.6 **	0.000	53.32
Block	16	5432	339.5	1.93 *	0.041	14.17
Interactions	21	16,082	765.8	4.35 **	0.000	41.95
IPCA 1	9	8448	938.6	5.34 **	0.000	52.53
IPCA 2	7	6175	882.1	5.02 **	0.000	38.40
Residuals	5	1460	291.9	1.66 ^ns^	0.162	3.81
Error	48	8441	175.9	*	*	

All sources were tested in relation to the error; **: Highly significant at *p* < 0.01 probability level; *: Significant at *p* < 0.05 probability level ^ns^: Not significant; Df: Degree of freedom; F: F value calculated; IPCA1: The first-interaction principal components axes; IPCA2: The second-interaction principal components axes.

**Table 6 plants-12-02165-t006:** First AMMI selections per environment.

No.	E	Mean	IPCA	Genotypic Rank			
		(cm)	Score	1	2	3	4
1	E1	241.5	4.671	G1	G4	G3	G2
4	E4	239.4	2.109	G1	G2	G4	G3
7	E7	225.0	1.204	G1	G4	G2	G3
8	E8	220.7	0.240	G2	G1	G4	G3
6	E6	224.0	0.128	G4	G3	G2	G1
5	E5	203.2	−1.712	G3	G4	G2	G1
3	E3	246.5	−2.943	G2	G1	G3	G4
2	E2	248.8	−3.698	G2	G1	G3	G4

E: Environments; IPCA score: Score based on the first-interaction principal component axes.

**Table 7 plants-12-02165-t007:** Average values for the kernel number per row of maize; interaction principal component values of IPCA1 and IPCA2 of the AMMI model; and the variance, standard deviation, and AMMI stability value (ASV) of four maize genotypes grown in eight different environments.

Parameter	Genotypes									
Environment	G1	G2	G3	G4	Em	IPCAe [*]	IPCAe [*]	Mean	Variance	SD
E1	30.90	26.28	28.22	29.82	28.81	−0.9884	0.2876	28.81	24.28	2.01
E2	37.29	33.93	33.75	34.65	34.91	−0.6269	−0.3823	34.91	3.68	1.64
E3	41.36	39.75	33.97	33.59	37.17	0.6796	−1.8670	37.17	23.75	3.97
E4	41.07	31.79	31.97	35.01	34.96	0.5706	0.7574	34.96	20.05	4.33
E5	27.95	23.63	28.04	29.82	27.36	−1.8007	0.7497	27.36	6.54	2.63
E6	36.60	34.63	32.30	32.45	33.99	−0.2808	−1.0859	33.99	5.85	2.04
E7	40.98	31.42	31.19	34.29	34.47	0.7541	0.7260	34.47	23.13	4.56
E8	38.98	27.20	25.49	29.19	30.22	1.6926	0.8147	30.22	32.02	6.04
IPCAg [*]	2.32563	0.24541	−1.5485	−1.0225						
IPCAg [*]	0.82715	−2.1785	0.01358	1.3378						
Gm	36.89	31.08	30.62	32.35						
ASV	3.6387	2.2104	2.3594	2.0535						

Gm: Genotype mean; Em: Environmental means; IPCAe1: The first-interaction principal component axes for environment; IPCAg1: The first-interaction principal component axes for genotype; IPCAe1: The second-interaction principal component axes for environment; IPCAg1: The second-interaction principal component axes for genotype; G: Maize genotypes; E: Environmental labels with control (K) and 70, 140, and 210 kg ha^−1^ nitrogen applied in both maize growing seasons; V: Variance; SD: Standard deviation; ASV: AMMI stability value.

**Table 8 plants-12-02165-t008:** AMMI analysis of variance for the kernel number per row of maize of four maize genotypes studied across eight environments.

Source ^1^	Df.	SS	MS	F-Value	F Prob.	The Share of Total Variation %
Total	95	2547.3	26.81	*	*	*
Treatments	31	2030.1	65.49	8.97 **	0.000	79.70
Genotypes	3	591.5	197.16	27.02 **	0.000	29.14
Environments	7	1015	145.01	13.91 **	0.000	50.00
Block	16	166.8	10.43	1.43 ^ns^	0.1687	8.22
Interactions	21	423.6	20.17	2.76 **	0.0018	20.87
IPCA 1	9	238.3	26.48	3.63 **	0.0016	56.26
IPCA 2	7	156.4	22.34	3.02 *	0.0095	36.92
Residuals	5	28.9	5.78	0.79 ^ns^	0.5604	1.42
Error	48	350.3	7.3	*	*	*

^1^ **: Highly significant at *p* < 0.01 probability level; *: Significant at *p* < 0.05 probability level; ^ns^: Not significant; Df: Degree of freedom; F: F value calculated; IPCA1: The first-interaction principal components axes; IPCA2: The second-interaction principal components axes.

**Table 9 plants-12-02165-t009:** The first AMMI selections per environment.

No.	E	Mean	IPCA	Genotypic Rank			
			Score	1	2	3	4
8	E8	30.22	1.6926	G1	G4	G2	G3
7	E7	34.47	0.754	G1	G4	G2	G3
3	E3	37.17	0.6795	G1	G2	G3	G4
4	E4	34.96	0.5706	G1	G4	G3	G2
6	E6	33.99	−0.2808	G1	G2	G4	G3
2	E2	34.91	−0.6269	G1	G4	G2	G3
1	E1	28.81	−0.9884	G1	G4	G3	G2
5	E5	27.36	−1.8007	G4	G3	G1	G2

E: Environments; IPCA score: Score based on the values of the first-interaction principal component.

**Table 10 plants-12-02165-t010:** Description of the eight different environments used to evaluate the four maize genotypes.

Environment	Growing Season	Treatments by Nitrogen (Granular Urea), kg N ha^−1^
E1	2021	0–control
E2	2021	70
E3	2021	140
E4	2021	210
E5	2022	0–control
E6	2022	70
E7	2022	140
E8	2022	210

**Table 11 plants-12-02165-t011:** Chemical analysis of chernozem soil.

Parameter	pH		CaCO_3_	Humus	Total N	Mineral N-NH_4_-N	P_2_O_5_	K_2_O
Depth, cm	KCl	H_2_O	%			mg kg^−1^ N	mg 100 g^−1^ Soil	
0–30	7.12	8.01	15.55	3.06	0.15	10.74	44.10	43.36

## Data Availability

Data are contained within the article.

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
