# Peer review of "Multivariate Interaction Analysis of Zea mays L. Genotypes Growth Productivity in Different Environmental Conditions"

_plants, 2023, doi:10.3390/plants12112165_

Round 1

Reviewer 1 Report

Manuscript "Multivariate Interaction Analysis of Productivity Zea mays L. Genotypes Grown in Different Environmental Conditions" is very interesting.

General comments:
Authors examined the response of four maize genotypes to various N conditions. Authors estimated the effects of the genotype, environment and their interaction on three important yield traits of maize genotypes using AMMI analysis. Authors selected genotype with stable reaction across each treatment.

One of the aims of the manuscript was to select genotypes with stable responses in each treatment. However, the Authors did not use any measure of stability to achieve this aim of the paper. See: https://doi.org/10.1007/s10681-022-02967-4, https://doi.org/10.1007/s10681-022-03070-4, https://doi.org/10.1007/s00217-021-03861-4

Detailed comments:
The results (Table 2) show that there are no statistically significant differences between genotypes. I don't see the point of conducting AMMI in the absence of differences between genotypes. Moreover, four genotypes is far too small a sample.

AMMI ANOVA showed that only IPCA1 was statistically significant. Which confirms the above that in this case AMMI is not the most accurate choice of method.

My suggestion:
Line 118: "[44]." not '[44"
"t ha" not "tha", throughout the manuscript.
"kg ha" not "kgha", throughout the manuscript.

Paper needs major revision.

Author Response

Dear appreciated Reviewer 1,

Thank you for your suggestions. We have adopted all your suggestions and thanks to the indicated suggestions, we have significantly improved our Manuscript Manuscript ID plants-2352729 , with the Title: “Multivariate Interaction Analysis of Productivity Zea mays L. Genotypes Grown in Different Environmental Conditions”, by the authors: Nataša Ljubičić , Vera Popovic * , Marko Kostić , Miloš Pajić, Maša Buđen , Kosta Gligorević , Milan Drazic , Milica Bižić , Vladimir Crnojevic.

Best regards,

Authors

Our answers to your questions are as follows.

Manuscript "Multivariate Interaction Analysis of Productivity Zea mays L. Genotypes Grown in Different Environmental Conditions" is very interesting.
General comments:
Authors examined the response of four maize genotypes to various N conditions. Authors estimated the effects of the genotype, environment and their interaction on three important yield traits of maize genotypes using AMMI analysis. Authors selected genotype with stable reaction across each treatment.

-Reviewer suggestion: One of the aims of the manuscript was to select genotypes with stable responses in each treatment. However, the Authors did not use any measure of stability to achieve this aim of the paper. See: https://doi.org/10.1007/s10681-022-02967-4, https://doi.org/10.1007/s10681-022-03070-4, https://doi.org/10.1007/s00217-021-03861-4

Author response:  In AMMI analysis, as an indicator of the stability of a genotype over environments the IPCA1 score of a genotype has been used. The IPCA value which poses long distance from zero indicates genotype instability, while zero IPCA value indicates the highest stability (The detailed explanation of model and stability has been presented in Materials and Methods)

Reviewer suggestion: The results (Table 2) show that there are no statistically significant differences between genotypes. I don't see the point of conducting AMMI in the absence of differences between genotypes.

Author response:

This results showed a large sum of squares for the environments (which was much higher than the share of the sum of squares of genotypes), indicated that the environments were diverse causing variation in the grain yield. A low sum of squares of genotypes could be a consequence of homogeneous genotypes in the study. Beside there were no statistically significant differences between genotypes, which appear there was no difference with them for analyzed trait, detailed separation of GEI variation which was significant at 5 %, it was observed that explainable variation of GEI had been carried out by the first IPCA1 axis in proportion of 62 % of total GEI variance.

If in a model are not selected any of significant IPCA, than AMMI model could not be the most accurate model. In this study, in table 2., GEI was found just significant (at 5 %), while detailed analysis of GEI showed that 74.59 % of this interaction was agronomical highly significant (at 1%) and defined with the first IPCA1.

Reviewer suggestion: Moreover, four genotypes is far too small a sample.

Author response: In the present study it was selected 4 maize genotypes because these 4 genotypes are one of the leading maize hybrids in agricultural production on farm production and therefore their estimation in different conditions is of great importance for genotype recommendation for maize production.
Reviewer suggestion: AMMI ANOVA showed that only IPCA1 was statistically significant. Which confirms the above that in this case AMMI is not the most accurate choice of method.

Author response: Genotype environment interaction (GEI) is integral source of variation, which involve genotype and the effect environmental factor. This source of variation (GEI) is needful, because selected genotype variation or environment variation is not enough to provide real information about genotype or environment. Using classical ANOVA as additive model, can be described only additive main effects and and nonadditive (GEI), but because of high value degree of freedom GEI can be non-significant, beside of great amount of sum of square, which exactly can be agronomical and statistical important. For that reason in AMMI model using PCA analysis genotype environment interaction (GEI) interaction can be separated.  All IPCA which are significant can explain GEI variation, while other IPCA which are not significant are added to the residuals.  If in a model are not selected any of significant IPCA, than AMMI model is not the most accurate model. In this study, in table 2., GEI was found just significant (at 5 %), while detailed analysis of GEI showed that 74.59 % of this interaction was agronomical highly significant (at 1%) and defined with the first IPCA1. The share of the first principal component (IPCA1) was highly significant at 1 %, which mean that The great variation (74,59 %) was brought up on the first PCA.

Other components which were not significant are considered as agonomically no significant and they are grouped in Residuals.

Row 118: Suggestion of reviewer 1: Line 118: "[44]." not '[44 Since that grain yield (missing bracket after number 44) Suggestion of reviewer 1: Line 118: "[44]." not '[44"

Correction: It is corrected and we input missing bracket on [44 ]. Since that grain yield

Comment and suggestion: "t ha" not "tha", throughout the manuscript.
"kg ha" not "kgha", throughout the manuscript.

Correction: Thank you for your suggestion. It is now corrected units on all places in manuscript.

With kind regard,

Authors

Reviewer 2 Report

The topic of this research is interesting and relevant to the main theme of the journal. Overall the manuscript is written well. However, the discussion seems to be weak. I suggest to include more recent references and provide a specific conclusion of this study. 

Minor changes are required at certain sections of the manuscript.

Author Response

Dear appreciated Reviewer 2,

Thank you for your suggestions. We have adopted all your suggestions and thanks to the indicated suggestions, we have significantly improved our Manuscript ID plants-2352729 , with the Title: “Multivariate Interaction Analysis of Productivity Zea mays L. Genotypes Grown in Different Environmental Conditions”, by the authors: Nataša Ljubičić , Vera Popovic * , Marko Kostić , Miloš Pajić, Maša Buđen , Kosta Gligorević , Milan Drazic , Milica Bižić , Vladimir Crnojevic.

Best regards,

Authors

Our answers to your questions are as follows.

The discussion and Conclusion was expanded and add new references.

  1. Discusion

Higher yields and biomass production of the hybrids with the latest genetic stock can be associated with increased plant nutrient uptake and higher utilisation of soil nutrient content [81]. Optimal nitrogen supply has a significant role in the growth characteristics of plants, as it is the main factor for plant cell components, primarily due to its role played in the photosynthetic apparatus [82]. The nitrogen use efficiency (NUE) of maize is estimated at 33% globally, which is negatively affected by fertiliser leaching under the root zone and denitrification [83]. According to Surendran et al. [84], maize absorbs about 10–20% of its total nitrogen requirements up to the V4 stage, whereas during 6 weeks of growth from V4 to VT, N accumulation approaches 60–70% of total N uptake. Growth stages have different nutrient demands based on their actual demand. Nitrogen and potassium had their maximum effect on the stalk of maize during the growing season. Magnesium and copper were the second most important and desirable factors during the different growth stages and treatments in relation to the stalk. Nitrogen and calcium had their maximum impact during the yield formation stage and nitrogen and phosphorus had their most desirable effect during the grain filling stage [85]. Total nitrogen was an essential factor of maize in all growth stages. Increased nitrogen fertilizer leads to more dry matter production and grain yield. Increasing nitrogen accelerates green growth, increases the above-ground mass of the plant, and increases the evaporation of plants and causes the roots to expand and bulk up [86]. Numerous reports recorded the positive effect of nitrogen on grain growth per grain, grain weight, and grain yield of maize, with a tendency to use higher amounts of nitrogen fertiliser [87–90].

  1. Conclusions

Important issues in using fertilizers is the comparison of methods and amounts of fertilizer use, which is very important from the aspect of increasing maize yield and economic production. The results of the investigation indicated that the grain yield, plant height and number kernel per row of examined maize genotypes were highly influenced by GEI effects. Additive Main effects and Multiplicative Interaction (AMMI) analysis showed a significant effect of the genotypes, environments and the effect of GEI for all traits. The significance of GEI interaction effect indicated that maize genotypes responded differently to various environment conditions. Varying fertilizer levels create environments and hybrids respond differently to the environments because of the differences in the genotypic constitution of the hybrids. Changing weather conditions additionally influenced the effect of nitrogen fertilizations and the observed variation in the grain yield traits of maize. The first growing season of investigation was followed with more favorable weather conditions for the maize crop from the aspects of precipitation, while the second growing season was dry in the critical stages of maize growth and it affected the values of the average yields over genotypes and treatments. More favorable weather conditions of the first growing season exhibited significant impact on achieving higher mean values of traits and reduce the differences between the mean values per treatment.

Based on the tested genotypes, genotype G3, beside of lower mean value than average mean and expressed the most stable reaction over all environments for the grain yield of maize. In this analysis, maize genotype G1 expresses the greatest value of grain yield, but followed with high interaction values, which indicates it as the least suitable for the stable establishment grain yield. 

Regarding to the trait the plant height, genotype G3, beside of lower mean value than average mean of this trait, expressed the most stable reaction over all environments for the trait plant height of maize.

Regarding to the trait the kernel number per row of maize, genotype G2, with lower mean value than average mean of this trait, expressed the most stable reaction over all environments.

This research revealed that the plant height, number kernel per row and grain yield of maize had important factors to achieve optimized fertilizer on the basis of sustainable agricultural on maize hybrids. The study showed variation for all studied traits among the maize genotypes, which indicates sufficient variability that can be exploited through selection, as well as for recommendation of suitable maize genotypes in examined environments. The effect of fertilizers on maize yield components and knowledge regarding their effect could assist to maintain balance and stability grain yield with application fertilizers. The stable maize hybrids could be recommended for both N stress and optimum nitrogen conditions. The maize hybrids identified for the various N conditions will support seed companies to make decision on the maize varieties to produce for farmer’s production. Therefore, the constraints to maize production due to soil fertility can be reduced. The effect of nitrogen on the quantitative properties of maize showed that nitrogen increases the production of grain yield and its components. Determination of the fertilization effect on the grain yield and partitioning of modern maize hybrids analyzed with different multivariate statistical methods can provide a basis for the farmers to implement hybrid and site-specific nutrient management systems with the goal of reducing the environmental impact of over fertilization and more rational production.

  1. New references

  1. Xin , Zhang J., Zhu A., Zhang C. Effects of long-term (23 years) mineral fertilizer and compost application on physical properties of fluvo-aquic soil in the North China Plain,” Soil and Tillage Research, 2016, 156: 166–172.
  2. Bender R., Haegele J.W., Ruffo M.L., Below F.E. Nutrient uptake, partitioning, and remobilization in modern, transgenic insect-protected maize hybrids, Agronomy Journal, 2013, 105, 1: 161–170.
  3. Pandey K., Maranville J.W., Admou A. Deficit irrigation and nitrogen effects on maize in a Sahelian environment, Agricultural Water Management, 2000, 46, 1: 1–13.
  4. Raun R., Johnson G.V. Improving nitrogen use efficiency for cereal production, Agronomy Journal, 1999, 91, 3: 357–363.
  5. Surendran U, Murugappan , Bhaskaran A., Jagadeeswaran R. Nutrient budgeting using NUTMON-Toolbox in an irrigated farm of semi arid tropical region in India-A micro and meso level modeling study,” World Journal of Agricultural Sciences, 2005, 1, 1: 89–97.
  6. Csaba B., Illés Á., Nasir Mousavi S. M., Széles A., Tóth B., Nagy J., Marton C.L. Evaluation of the Nutrient Composition of Maize in Different NPK Fertilizer Levels Based on Multivariate Method Analysis. International Journal of Agronomy. 2021, ID 5537549, https://doi.org/10.1155/2021/5537549
  7. Ali I., Dawelbeit S.E., Salih A.A. Effect of Water Stress and Nitrogen Application on Grain Yield of Wheat, Agricultural Research and Technology Corporation Unit, New Delhi, India, 2006.
  8. Asghar , Muhammad W., Asif T., Muhammad T., Nadeem M.A., Zamir M.S.I. Impact of nitrogen application on growth and yield of maize (Zea mays L.) grown alone and in combination with cowpea (Vigna unguiculata L.),American-eurasian Journal of Agricultural and Environmental Science, 2010, 7, 1, 43–47.
  9. Božović , Popović V., Rajičić V., Kostić M., Filipović M., Kolarić Lj., Ugrenović V, Spalević V. Stability of the expression of the maize productivity parameters by AMMI models and GGE-biplot analysis, Notulae Botanicae Horti Agrobotanici Cluj-Napoca, 2020, 48, 3: 1387–1397.
  10. Sharifi R. S., Taghizadeh Response of maize (Zea mays L.) cultivars to different levels of nitrogen fertilizer, Journal of Food, Agriculture & Environment, 2009, 7, 3/4: 518–521.

Reviewer 3 Report

The manuscript in its current form is weak in language; a lot of editing and rectification of grammatical errors is to be done. I suggest taking editorial assistance from language experts. As well as the rational of the problem suggested in the attached file seems to be weak, needs substantial improvement. This manuscript is totally dependent on the weather data; please carefully check the data and fix the suggested issue. There is a big issue while using the abbreviations. Abbreviations should be elaborated where first mentioned in the manuscript. Not write full forms again and again. Rectify in the entire manuscript. Units should be according to SI.

The manuscript in its current form is weak in language; a lot of editing and rectification of grammatical errors is to be done.

Author Response

Dear appreciated Reviewer 3,

Thank you for your suggestions. We have adopted all your suggestions and thanks to the indicated suggestions, we have significantly improved our Manuscript Manuscript ID plants-2352729 , with the Title: “Multivariate Interaction Analysis of Productivity Zea mays L. Genotypes Grown in Different Environmental Conditions”, by the authors:  Nataša Ljubičić , Vera Popovic * , Marko Kostić , Miloš Pajić, Maša Buđen , Kosta Gligorević , Milan Drazic , Milica Bižić , Vladimir Crnojevic.

Best regards,

Authors

Our answers to your questions are as follows.

According to the suggestion in Manuscript, the suggested correction was done (below):

Abstract

Row 19-20: Across two growing seasons, phenotypic variability and GEI for yield traits over four maize genotypes grown in four different (should be mentioned word here) environments were studied.

Corrected: Across two growing seasons, phenotypic variability and GEI for yield traits over four maize genotypes grown in four different fertilization environments were studied.

Introduction

Row 47: nitrogen (N) it was wrote to be consisted in abbreviations

Corrected: According to your suggestion, “nitrogen (N)” is only on first mention remarked with both full name and abbreviation and in further text is consistent.

Row 51-52: Climatic conditions and nitrogen (N) fertilizers applica-51
tion significantly affect the maize yield (Zea mays L.) productivity – there is no nee too put scientific name of maize

Corrected: According to your suggestion: Climatic conditions and nitrogen fertilizers application significantly affect the maize yield productivity.

Row 58: Despite the quantity of maize yield, as well as, balanced use of nitrogen fertilizers is one of the main goals to maize breeders is to create maize genotypes with high grain yield with the stable reaction
in different environments, as well as in conditions of low soil fertility and less fertilization supply.

Corrected: Despite the quantity of maize yield, as well as, balanced use of nitrogen fertilizers is one of the main goals to maize breeders is to produce maize genotypes with high grain yield stable reaction in different environments, as well as in conditions of low soil fertility and less fertilization supply (word “create” is changed to produce; “with” is deleted)

Row 62 and 63 and 72: genotype by environment interaction (GEI) id consistent (without other style)

Row 77:  It is delete one more space between these 2 words GEI), [21].

Row 87 and 91: Additive Main Effect and Multiplicative Interaction model it is corrected on both rows to be consistent

Correction: Additive main effect and multiplicative interaction model

Row 93: It is deleted repetition of abbreviation with full name.

Correction: In the AMMI model analysis of variance serve to divide the variation into genotype main effects (G), environment main effects (E) and genotype by environment interaction (GEI) and afterwards, it applies principal components analysis (PCA), to analyze the residual GEI effect, [20, 25].

Row 106: yield    traits

Correction: It is deleted one more space between 2 words.

  1. Results and Discussion

Row 118: [44 Since that grain (missing bracket after number 44)

Correction: It is corrected on “ [44 ]. Since that grain”

Row 207 and row 209, as well as in other parts of paper, I corrected mistake which refers on nitrogen amount. There was mistake in some places and instead of treatment expressed in of kg N ha-1 for environments it was wrote tha−1 nitrogen application - mistake

Correction: Thank you for seeing this mistake in writing, on each place for treatment is now corrected on:

 kg N ha-1

  1. Materials and Method

Rows from 693 to 696 was repetition of similar sentence:

In the trial the first treatment was soil without nitrogen application (control), the second treatment was nitrogen application using granular urea in the amount of 70 tha−1, the third treatment was nitrogen application using granular urea in the amount of 140
tha−1 and fourth treatment was nitrogen application using granular urea in the amount
of 200 tha−1.  (Repetition)

Correction: This sentence is deleted according to the suggestions, since it was repetition

Row 717: In table 6 was mistake in writing amount of applied fertilizer in measure units

Correction: On each place it is now corrected nitrogen amount in kg ha-1 (instead of tha-1)

3.3. Meteorological Conditions

Row 754-755:

The authors suggested to correct this part, because in text weren’t pointed out that was enough precipitation, but it wasn’t pointed out that was unfavoite schedule of rain amount. It was missing in critical stage of maize development.

“Average temperatures in the growing season in 2022 were 19.46 °C and were higher753
by 0.98 °C compared to growing season of 2021, while total precipitation in 2022 was754
higher by 31.6 mm.”

Corrected: Average temperatures in the growing season in 2022 were 19.46 °C and were higher by 0.98 °C compared to growing season of 2021, while total precipitation in 2022 was higher by 31.6 mm, but with an unfavorable schedule, specially in critical stages of maize (Thank you it is corrected now)

  1. Conclusion

Row 806 is now in accordance with weather data

Row 844-845: Author Contributions: For research articles with several authors, a short paragraph specifying their individual contributions must be provided. he following statements should be used “Conceptualization, V.P., V.C. and N.Lj.; methodology, N.LJ.; software, M.K. and M.P.; validation, V.P., M.B. and N.LJ.; formal analysis, V.P., N.Lj.; investigation, V.P.; resources, V.P.; data curation, M.B.; writing—original draft preparation, N.LJ.; V.P; K.G. and M,D.; writing—review and editing, M.B., V.C.; N.LJ. and V.P.; visualization, V.C.; supervision, V.P.; project administration, V.P. and N.Lj. All authors have read and agreed to the published version of the manuscript.” (red part of text wasn’t deleted)

Corrected: Conceptualization, V.P., V.C. and N.Lj.; methodology, N.LJ.; software, M.K. and M.P.; validation, V.P., M.B. and N.LJ.; formal analysis, V.P., N.Lj.; investigation, V.P.; resources, V.P.; data curation, M.B.; writing—original draft preparation, N.LJ.; V.P; K.G. and M,D.; writing—review and editing, M.B., V.C.; N.LJ. and V.P.; visualization, V.C.; supervision, V.P.; project administration, V.P. and N.Lj. All authors have read and agreed to the published version of the manuscript.”

The manuscript in its current form is weak in language; a lot of editing and rectification of grammatical errors is to be done.  I suggest taking editorial assistance from language experts. As well as the rational of the problem suggested in the attached file seems to be weak, needs substantial improvement.

Reviewer suggestion: This manuscript is totally dependent on the weather data; please carefully check the data and fix the suggested issue

Corrected:Thank you for your valuable suggestions, we made corrections and extended and make more clear  part regarding to the weather data.

Reviewer suggestion: There is a big issue while using the abbreviations. Abbreviations should be elaborated where first mentioned in the manuscript. Not write full forms again and again. Rectify in the entire manuscript.

Corrected: Dear appreciated reviewers, we made corrections and put appropriate units according to the SI.

Best regards,

Dr. Nataša Ljubičić and authors

Round 2

Reviewer 1 Report

Considering only the value of IPCA1 jeko genotype stability is not enough! A standard measure of stability using the AMMI model is the AMMI stability value (ASV) [Purchase et al., 2000]. See: https://doi.org/10.1007/s10681-022-02967-4, https://doi.org/10.1007/s10681-022-03070-4, https://doi.org/10.1007/s00217-021-03861-4

Zróżnicowanie środowisk powoduje zmienność plonu ziarna. Tak, zgadza się. Jednakże przy braku różnic między genotypami oznacza to, że genotypy plonują podobnie w danym środowisku. A różnice między środowiskami są jedynie wprost proporcjonalne. Brak różnic między genotypami oznacza, że w danym środowisku genotypy są tak samo stabilne.
Istotność GEI wynika z bardzo dużych różnic między środowiskami. Zastosowanie miary ASV wskaże, że wszystkie genotypy są jednakowo stabilne.

Moreover, four genotypes is far too small a sample.! It is not true that only these four corn genotypes meet the requirements of the authors. Many researchers have a very large collection of such genotypes. AMMI model for four genotypes does not make sense!

The authors' explanations to my comments confirm that using the AMMI model for the data presented is not an accurate choice. A better one would be to use REML analysis.

Paper needs major revision.

Author Response

Dear appreciated reviewer,

Thank you for the proposed corrections and useful suggestions to make the work as valuable as possible. According to your suggestions, we have made changes and additions in the paper.

Reviewer comment: Considering only the value of IPCA1 jeko genotype stability is not enough! A standard measure of stability using the AMMI model is the AMMI stability value (ASV) [Purchase et al., 2000]. See: https://doi.org/10.1007/s10681-022-02967-4,https://doi.org/10.1007/s10681-022-03070-4, https://doi.org/10.1007/s00217-021-03861-4

Author response:

Thank you for useful suggestion and valuable improvement which it made. We added in Manuscript beside of AMMI analysis, AMMI Stability Value (ASV) proposed by Purchase JL.

It gave interesting result. For all studied traits, the AMMI analysis and ASV ranking were in accordance and showed quite similar results. Thank you.

Reviewer comment: Moreover, four genotypes is far too small a sample.! It is not true that only these four corn genotypes meet the requirements of the authors. Many researchers have a very large collection of such genotypes. AMMI model for four genotypes does not make sense! The authors' explanations to my comments confirm that using the AMMI model for the data presented is not an accurate choice. A better one would be to use REML analysis.

Author response:

Thank you for useful suggestion. I agree that could be much higher number of maize genotypes especially in Breeding Institutes, Breeding companies with different breeding collection. I agree that numerous maize genotypes are present on our market and share with different maturity, different origin and different availability. Four different hybrids is enough for investigations and for this analysis, especially since in our study we adapted investigation with available hybrids with farmers which gave guarantee that will be the same genotypes repeated in seeding more than one growing season, to avoid changes in repeatability.  Using another analysis new analyzes change the flow and framework of work.

The work was completed with the requested parameters of stability, where the ASV values and standard deviation were entered for each parameter, as well as the comments, which agreed very well. In the paper, all proposed changes are marked in gray.

I would like to thank the respected reviewer for his suggestions. Thanks to your suggestions, we improved the work significantly and I hope that both sides will be satisfied.

Kind regards,

Authors

Reviewer 3 Report

The authors did not respond to my previous comments correctly. Please carefully revise the article and resubmit it... Still, the manuscript in its current form is weak in language; a lot of editing and rectification of grammatical errors is to be done. I suggest taking editorial assistance from language experts. Further comments are in the attached file.

Very poor in its current form.

Author Response

Thank you very much for the useful suggestion. Attached is the corrected paper. We adopted the reviewer's suggestions.  According to suggestions, we have made changes and additions in the paper.

 Reviewer comment: The authors did not respond to my previous comments correctly. Please carefully revise the article and resubmit it... Still, the manuscript in its current form is weak in language; a lot of editing and rectification of grammatical errors is to be done. I suggest taking editorial assistance from language experts. Further comments are in the attached file.

Author response:

Thank you for the useful suggestions and valuable improvement. We corrected all remarked details.

In Abstract in line 19 we added the names of genotypes used in the study.

At the same row the word environment is changed by word treatment.

Row 21:  Results of AMMI analyses revealed – the part of sentence “of AMMI analyses” is deleted.

Row 25: Space between the number and %, is deleted (74.6 %).

In Introduction part abbreviated word Nitrogen (N) is corrected and uniform now in the text.

Row 54: On the other side, sufficient amounts of fertilizer can cause environmental pollution.

Corrected: On the other side, application of sufficient N fertilizer amounts can cause environmental pollution.

Row 70: Maize genotypes have different reactions in different years, treatments, as well as in the combination of these factors, due to genotype by environment interaction (GEI).

Correction: Maize genotypes have different reactions in different years, treatments, as well as in the combination of these factors, due to GEI.

I would like to thank the respected reviewer for his suggestions. Thanks to your suggestions, we improved the work significantly and I hope that it is at all satisfied requiremnts.

Kind regards,

Authors

Round 3

Reviewer 1 Report

Now all is correct, except for the formula for the ASV. The power should be after the brackets, not inside.